# Individual Cofactors and Multisensory Contributions to the Postural Sway of Adults with Diabetes

**DOI:** 10.3390/brainsci12111489

**Published:** 2022-11-02

**Authors:** Julio César Villaseñor-Moreno, Catalina Aranda-Moreno, Ignacio Figueroa-Padilla, María Esther Giraldez-Fernández, Michael A. Gresty, Kathrine Jáuregui-Renaud

**Affiliations:** 1Unidad de Investigación Médica en Otoneurología, Instituto Mexicano del Seguro Social, Ciudad de Mexico 06720, Mexico; 2Hospital Regional 72, Instituto Mexicano del Seguro Social, Tlalnepantla de Baz 54000, Mexico; 3Hospital General de Zona 1A, Instituto Mexicano del Seguro Social, Ciudad de Mexico 03300, Mexico; 4Neuro-otology Unit, Imperial College London, Charing Cross Hospital, London W6 8RF, UK

**Keywords:** postural sway, type 2 diabetes, vestibular function, polyneuropathy, polypharmacy

## Abstract

To assess the interactions between individual cofactors and multisensory inputs on the postural sway of adults with type 2 diabetes and healthy subjects, 69 adults accepted to participate in the study (48 with/21 without diabetes). Assessments included neuro-otology (sinusoidal-rotation and unilateral-centrifugation), ophthalmology and physiatry evaluations, body mass index (BMI), physical activity, quadriceps strength, the ankle/brachial index and polypharmacy. Postural sway was recorded on hard/soft surface, either with eyes open/closed, or without/with 30° neck extension. The proportional differences from the baseline of each condition were analyzed using Multivariate and Multivariable analyses. Patients with polyneuropathy and no retinopathy showed visual dependence, while those with polyneuropathy and retinopathy showed adaptation. Across sensory challenges, the vestibulo-ocular gain at 1.28 Hz and the BMI were mainly related to changes in sway area, while the dynamic visual vertical was mainly related to changes in sway length. The ankle/brachial index was related to the effect of neck extension, with contributions from quadriceps strength/physical activity, polyneuropathy and polypharmacy. Across conditions, men showed less sway than women did. In conclusion, in adults with diabetes, sensory inputs and individual cofactors differently contribute to postural stability according to context. Rehabilitation programs for adults with diabetes may require an individualized approach.

## 1. Introduction

To avoid falling, postural control maintains the body’s center of mass within the stability area of the base of support (for review see [1]). This involves interactions among external constrains (e.g., gravity), the properties of the body and the neuromuscular forces (for review see [2]). The neuromuscular control adjusts continuously, according to the inputs from the somatosensory, visual and vestibular systems [3,4,5]; however, postural control may also vary according to individual cofactors, such as age and sex [6,7].

Type 2 diabetes (diabetes) is a major cause of both multisensory deficits and health conditions that can influence postural control [8,9,10,11,12,13,14,15,16]. Diabetes is a major cause of vision impairment and peripheral neuropathy [8], which decrease postural stability [9,10]; while a combined deficit of somatosensory and vestibular inputs can interfere with balance [11]. Diabetes is also associated with increased body mass index (BMI) [12], which may adversely affect postural stability [13,14,15], while it can be improved by loss of extra-weight [16]; furthermore, in patients with diabetes, a BMI ≥ 35 may increase fall risk [17]. Simultaneously, diabetes is associated with accelerated loss of muscle strength, volume, and quality over time [18]; nonetheless, diabetes promotes peripheral artery disease [19] that may contribute to balance impairment [20]. In addition, the medication of patients with diabetes frequently requires ≥5 drugs per day (polypharmacy) [21], which can interfere with postural control by their side effects [22,23]. Contrariwise, the practice of physical activity [24] and strong lower-extremity muscles [25] can improve balance. However, to the best of our knowledge, there are no reports on the combined effect that all these factors could have on the postural control of patients with diabetes.

To assess postural control, static posturography tracks the movement of the center of pressure at the feet of a standing subject, by means of several parameters (for review see [26,27]). In the time domain, the length of sway or sway path shows the displacement of the center of pressure during the recording time, and it has the greatest sensitivity for detecting changes in body sway with the less inter-subject variability [28], while the area of sway represents the portion of the base of support utilized during quiet stance, and it is a clinical predictor of falls [29,30]. The mean velocity is related to the frequency of the excursions of the center of pressure; however, when the recording time is constant (as it is in this study) the mean velocity is equivalent to the length of sway [26]. 

The relative contribution of the sensory systems to postural control can be assessed by changing the test conditions for each recording [31,32,33,34,35]: closing the eyes and reducing foot sensation by a compliant surface allow assessment of the visual and the somatosensory inputs, respectively [32], whereas, neck extension modifies the position of the utricles with respect to gravity, while it stimulates neck proprioceptors [33,34], and may also modify the blood flow in the vertebral arteries [35]. 

The purpose of this study was to explore the combined contributions of multisensory inputs and individual cofactors on the postural sway of patients with diabetes. Accordingly, we conducted a correlational study comprising somatosensory, visual, and vestibular evaluations, as well as BMI, physical activity, quadriceps isometric strength, the ankle/brachial index, polypharmacy and sex as cofactors. To perform the analysis on a wide range of responses, a subgroup of healthy subjects was also included in the study.

## 2. Materials and Methods

### 2.1. Participants

After approval by the Research and Ethics Committees of the Institution, 69 subjects participated in the study. They were invited to participate at the first level of medical care, within the same Institution. According to the selection criteria, none of them were seeking care due to sensory impairment or balance disorders; none had medical record or complains of otology, neurology, psychiatry, or orthopedic disorders, foot injuries, peripheral artery disease, severe renal failure, postural hypotension, or exposure to ototoxic medication or injurious noise levels. The participants were: Forty-eight patients with type 2 diabetes (mean age 58.0 ± standard deviation 11.2 years, 37 women/11 men), with a mean BMI of 28.9 ± 4.5, among whom eighteen (37.5%) had high blood pressure, which was controlled by medication.Twenty-one healthy subjects (age 51.3 ± 8.2 years, 13 women/8 men), with a BMI of 28.4 ± 4.2, among whom three (14.3%) had high blood pressure, which was controlled by medication.

A required sample size of 50 subjects was considered in order to perform an analysis of covariance when *R*^2^ = 0.1 for the tested variable and *R*^2^ = 0.3 for the controlled variables, with *α*= 0.05 and *β* = 0.2 [36]. In addition to the participants listed above, three more patients with similar characteristics to the whole group accepted to participate in the study, but failed to attend all the appointments, due to personal reasons that were not related to the study.

### 2.2. Procedures

After self-administration of an in-house medical questionnaire (including medication) and the short form of the International Physical Activity Questionnaire [37] (Version 2.0 Guidelines, www.ipaq.ki.se, accessed on 4 October 2022), the following assessments were performed:Vision was assessed by an ophthalmologist, with retinoscopy.Peripheral neuropathy was assessed by the Michigan Diabetic Neuropathy Score and Semmes-Weinstein 10 g monofilament test. When any of these instruments was positive, the tibial and sural nerves conduction was evaluated [38,39] (Spirit, Nicolet Instrument Corporation, Madison, WI, USA).Vestibular assessment included both the angular vestibulo-ocular responses (VOR) to sinusoidal rotations (0.16 Hz and 1.28 Hz, 60°/s peak) and the dynamic subjective visual vertical (DVV) during right/left centrifugation for gravity perception [40] (I-Portal-NOCT-Professional, Neuro-Kinetics, Pittsburgh, PA, USA). According to the protocol of the manufacturer, unilateral centrifugation was performed at 300°/s peak velocity/3.85 cm linear translation to the right and to the left; with 2 s to start dwell, 60 s ramp-up time, 375 s peak time, 60 s centrifugation start time and 60 s ramp down time. Subjects were instructed to set a laser line to the vertical position where they were offset 3.85 cm to the right or left of the center of rotation during full speed rotation [40], the mean deviation for each position was used for the analysis.The body weight and height were measured to estimate the BMI. Obesity was considered when BMI ≥ 30 kg/m^2^.Peripheral artery function was evaluated by the ankle/brachial index (Homecare JPD-100b+, Shenzhen Jumper Medical Equipment Co, Shenzhen, China). This index is a clinical metric for arterial stiffness [41].Lower-extremity muscle strength was estimated by quadriceps isometric strength (average of 5 right and 5 left attempts) (Baseline, Back-leg-chest dynamometer, White Plains, NY, USA), which has been related to both hyperglycemia and peripheral neuropathy [42].Static posturography was performed using a force platform (900 points/kg resolution; 40 Hz sampling rate; 16 b. analogue-digital converter) (Posturolab 40/16, Medicapteurs, Balma, France). Subjects were asked to stand upright and barefoot on the platform as still as possible, with arms at their sides. According to the manufacturer reference, feet position was maintained to record each baseline and its corresponding sensory challenge (30° lateral rotation between the feet and heels 3.5 cm apart). Recordings were made during 51.2 s [27], either with eyes open/closed, with/without 30° neck extension, and while adding or not a layer of foam rubber (5 cm thick, density of 2.5 pounds per cubic foot) to the base of support. The order of the eight data sets to perform the four sensory challenges included in the study is described in Table 1. To take into account that the subjective perception of the task’s difficulty may lead to sway increase [43], all the participants were exposed from a condition similar to daily life activities (i.e., standing with the eyes open on a hard surface) to the most unusual condition (i.e., standing with the eyes open on a soft surface with neck extension). In order to identify the response to each sensory challenge, we calculated the proportional difference between each of the baselines and the corresponding consecutive recording (i.e., 1 and 2, 3 and 4, 5 and 6, and 7 and 8), which was considered as an ‘effect’ measure on the length of sway (mm) and the 90% confidence ellipse of the area of sway (mm^2^), which were obtained by the software provided by the manufacturer of the platform (Medicapteurs, Balma, France) (for review see [44]).

### 2.3. Statistical Analysis

After assessment of data distribution using Kolmogorov-Smirnov test, an exploratory bivariate analysis was performed using “*t*” test (either for means or for proportions), Mann Whitney U test and Pearson’s linear correlation. In order to assess the multisensory contributions to sway, we contrasted the responses to each sensory challenge by repeated measures analysis, using multivariate analysis of covariance (MANCOVA), taking into account the individual cofactors. Subsequently, we also analyzed linear and non-linear relationships among the study variables for each sensory challenge by multivariable analysis (Generalized Linear Model with Wald test), selecting the models according to likelihood ratios, in order to improve accuracy of effect estimates [45]. All tests were performed using a two tailed significance level of 0.05.

## 3. Results

### 3.1. Characteristics of the Participants

In patients with diabetes, the frequency of both polyneuropathy and retinopathy was circa 35%, while a fifth of them had combined polyneuropathy and retinopathy (Table 2). Fifteen (31.2%) patients were using ≥5 drugs per day; the most frequent medication were metformin (58%) and insulin (34%), which were frequently combined with high blood pressure medication (34%), of which the most frequent was losartan (23%), while hydrochlorothiazide was used only by 5 (10%) patients. 

Comparisons between the group of patients with diabetes and the group of healthy subjects showed that the proportion of women was similar in the two groups (*p* = 0.20), while the patients were older than the healthy subjects (“*t*” test, *t* = 2.482, *p* = 0.01). 

As could be expected, the mean quadriceps strength and the score on physical activity were lower in patients than in healthy subjects (Table 2). Although the mean ankle brachial index was similar in the two groups, it was more variable among patients than among healthy subjects. Similarly, the VOR gain and the mean DVV were similar in the two groups, but they were also more variable among patients than healthy subjects (Table 2). Of note, the individual results showed that six of the patients perceived no deviation of the DVV during any of the right and left centrifugations (bilateral dysfunction), while 11 of them perceived deviation just during one of the two centrifugations, suggesting unilateral otolithic dysfunction [40]. No abnormalities were observed in the VOR to sinusoidal rotation in the dark.

The correlation matrix of the variables described above showed low to moderate correlations on the quadriceps strength, which was related to the age of the participants (*r* = 0.24, *p* = 0.04), to the physical activity (*r* = 0.48, *p* < 0.001), to the ankle/brachial index (*r* = 0.26, *p* = 0.03) and to the DVV (*r* = 0.26, *p* = 0.02). Interestingly this was the only significant correlation for the age of the participants.

### 3.2. Bivariate Analyses on Postural Sway

Comparisons between men and women showed that, during each of the eight recordings, men had 16% to 27% shorter length of sway than women (Mann Whitney *U* test, *Z* = −3.95 to −2.69, *p* < 0.006); and, during recordings 1, 4, 6 and 8, men had 17% to 32% smaller area of sway than women (Mann Whitney *U* test, *Z* = −2.02 to −2.65, *p* < 0.05).

Comparisons between patients and healthy subjects showed that after closing the eyes while standing on hard surface, patients showed a larger effect (proportional difference) in the area of sway than healthy subjects (Mann Whitney *U* test, *Z* = 3.28 *p* = 0.0007) (Figure 1), with no difference in the length of sway.

### 3.3. Repeated Measures Analysis among the Sensory Challenges

Figure 1 shows the effects in sway for all the sensory challenges. Closing the eyes had the largest effect on sway in both patients and healthy subjects (MANCOVA, *p* < 0.01), while neck extension had the least effect on sway, even while standing on soft surface.

Closing the eyes while standing on soft surface provoked a larger increase in the area of sway in patients with polyneuropathy and no retinopathy than in those with both polyneuropathy and retinopathy (MANCOVA, *F* = 2.641, *p* = 0.051) (Figure 2). In addition, during this condition, men with polyneuropathy showed a larger increase in the area of sway than women (MANCOVA, *F* = 3.823, *p* = 0.01) (Figure 3).

After closing the eyes while standing on soft surface, patients with polyneuropathy and no polypharmacy also showed a larger increase in the length of sway than patients with polyneuropathy and polypharmacy (Figure 4). However, an opposite effect was observed after 30° neck extension, when patients with polyneuropathy and polypharmacy showed a larger increase in the length of sway than those with polyneuropathy and no polypharmacy (MANCOVA, *F* = 6.548, *p* = 0.003) (Figure 4).

### 3.4. Multivariable Analysis of Each Sensory Challenge

During the first baseline recording (eyes open/hard surface), the variables related to the length of sway were a BMI ≥ 30 and sex (Wald Statistic ≥ 6.2, *p* < 0.02) (Table 3). The variables related to the area of sway were the VOR gain to sinusoidal rotation at 1.28 Hz, a BMI ≥ 30 and the quadriceps strength (Wald Statistic ≥ 4.6, *p* < 0.05) (Table 3). Results of a similar analysis on the effect of each sensory challenge were as follows:

#### 3.4.1. Postural Sway after Closing the Eyes While Standing Either on Hard or Soft Surface

Hard surface. The variables related to the effect in the length of sway were the age and the VOR gain at 1.28 Hz (Wald statistic ≥ 3.87, *p* < 0.05); in the area of sway the variables were the BMI, the VOR gain to rotation at 1.28 Hz, the average DVV, physical activity, polyneuropathy and an additive contribution of polyneuropathy and polypharmacy (Wald statistic ≥ 4.07, *p* < 0.05).Soft surface. The variables related to the effect in the length of sway were the VOR gain at 1.28 Hz and the average DVV (Wald statistic ≥ 4.33, *p* < 0.05); in the area of sway the variables were the BMI, physical activity and quadriceps strength (Wald statistic ≥ 3.86, *p* < 0.05).

#### 3.4.2. Postural Sway after 30° Neck Extension While Standing Either on Hard or Soft Surface

Hard surface. The variables related to the effect in the length of sway were the average DVV, the quadriceps strength and the ankle/brachial index, with an additive effect of polyneuropathy and polypharmacy that was observed just in men (Wald statistic ≥ 4.61, *p* < 0.05). In the area of sway the variables were the BMI, the VOR gain at 1.28 Hz, and polyneuropathy (particularly in men) (Wald statistic ≥ 4.38, *p* < 0.05).Soft surface. The variables related to the effect in the length of sway were the DVV, the physical activity score, the quadriceps strength, the ankle/brachial index and polypharmacy (Wald statistic ≥ 4.23, *p* < 0.05); in the area of sway the variable was the quadriceps strength, with a larger increase in the area of sway in patients with retinopathy and no polyneuropathy (Wald statistic ≥ 4.26, *p* < 0.05).

## 4. Discussion

Individual cofactors contributed differently to sway changes according to the sensory challenges.

### 4.1. Multisensory Contributions

The length of sway was greatest with eye closure while standing on a soft surface. This finding is consistent with the evidence that vision provides a substantial contribution to postural stability; whereas, increased sway with polyneuropathy indicated the important role of haptic senses in posture [32]. Interestingly, polypharmacy and retinopathy reduced the effect of eye closure presumably because subjects with retinopathy had adapted to reduced visual input to balance, while, for this sensory challenge, medication might have ameliorated destabilizing influences to some extent. 

The angular VOR gain to sinusoidal rotation in the dark at 1.28 Hz was related to postural sway over a variety of sensory challenges; however, this relationship was not evident for rotation at 0.16 Hz. These results are consistent with the evidence supporting that vestibular input may be a contributing factor to modulate sway in the middle frequency range. Previous studies have shown that the majority of energy of postural sway lies within the bandwidth of 0.0–2.0 Hz, including clustering of sway that may be related to the effects of physiological events such as breathing and heartbeat [46]. More detailed evidence support that frequencies between 0.3 and 0.8 Hz may be linked to somatosensory inputs, while vision may affect the spectral content in nearly all frequencies in the anterior-posterior direction and both frequencies ≤1 Hz and >3 Hz in the medial-lateral direction [47]. Furthermore, peripheral vestibular disease has been related to sway in a frequency band from 0.1 to 1 Hz, when visual and somatosensory inputs are reduced (by closing the eyes while standing on soft surface) [48]. 

We also observed a relationship between the average DVV and the effect of closing the eyes while standing on hard surface. This result is consistent with a previous report of a strong correlation between the torsional eye movement response to roll head-tilt and the area of sway, while standing on a firm surface [49]. Additionally, we observed that the average DVV was related to the proportional difference in the length of sway after 30° neck extension (while standing either on hard or soft surface), with contribution from the quadriceps strength; therefore the amount of sway was influenced by both the perception of verticality and the muscular strength of the legs. It has been proposed that head-tilt may decrease the ability of the vestibular system to discern the orientation of upright [50]. However, evidence has shown that, after 30° neck extension, in healthy adults and patients with vestibular disease, the amount of sway increase may be similar, with no additional effect while standing on soft surface than on a hard surface [33]. 

### 4.2. Individual Cofactors

The BMI influence on postural sway could be at least partially explained by inaccurate performance of motor corrections. Even in the absence of diabetes, increased BMI may convey fat infiltration in leg skeletal muscles [51], as well as peripheral nerve impairment [52]. Consequently, ankle strength and ankle movement perception may decline, with impaired physical performance [53,54]. Additionally, evidence suggest that subjects with obesity may use somatosensation to control posture differently than lean and overweight subjects [13,16].

The influence of muscle strength on postural sway was particularly evident in the most challenging situations. In order to keep the center of pressure within the limits of stability, subjects required additional muscular activity while standing on soft surface, and those with less strength had more sway. These findings were further corroborated by similar results on the report of physical activity. Moreover, muscle strength was related to age. 

In young adults, during upright stance, both ankle range of motion and muscle strength of the ankle, knee and hip (except hip extensor) may influence balance control [55]. Aging is related to decline in muscle strength [56,57], with decreased hallux plantar-flexion that is more pronounced in women than in men [58]. During upright stance on a soft surface with the eyes closed, older subjects may sway more to maintain the center of pressure within a similar area (length as a function of the area of sway) than younger subjects, increasing energy expenditure for the same task [13]. In addition to muscle weakness, patients with diabetes may also have increased ankle stiffness [59]. 

In this study we also observed an independent contribution of the ankle/brachial index on the proportional difference in the length of sway after 30° neck extension (either on hard or on soft surface). This finding suggests that underling artery stiffness may have contributed to the amount of sway when the neck was extended, which is consistent with the evidence of decrease of the blood flow velocity in the vertebral arteries during extension and extension-rotation of the head in healthy, young subjects [60]. Furthermore, evidence supports that borderline and low normal ankle-brachial index values can be associated with reduced thigh muscle energy production in adults without evidence of cardiovascular disease [61].

In this study, polypharmacy was related to sway mainly when sensory inputs were reduced. Sway increase was observed in patients with polypharmacy after 30° neck extension, while standing on a soft surface; however, after closing the eyes without neck extension, the effect was the opposite, and medication contributed to less increase in sway. The need of medication for diabetes was of course related to the control of diabetes, while the need of medication for high blood pressure implied cardiovascular disease; of note, patients were not taking psychotropics, and just a few of them were taking diuretics. Then the underlying disease may have also contributed to the observed effects. In the elderly, polypharmacy increases postural sway and the risk of falling [21,22], particularly because the probability of receiving a drug that may induce postural instability among the number of drugs taken (such as diuretics, antiarrhythmics and psychotropics) [22].

### 4.3. Limitations

Our results should be interpreted in the context of the limitations of the study. Since patients were invited to participate at primary health care, the results may be different in patients with more physical decline. Since most of our patients were middle-aged adults, the influence of age on sway was not directly evident, but mainly observed in the interactions among individual factors. The main technical limitation was the limited parameters used to represent a complex phenomenon; among the missing variables, we did not measure the ankle range of motion, neither the ankle dorsiflexor/plantarflexor strength, which may have contributed to the variance of sway. However, the strengths of the study include the prospective collection of the data by a team of trained health professionals and the selection of the participants, reducing the influence of severe diabetes complications. The scarcity of studies including the simultaneous evaluation and analysis of all the variables that were included in this study exposed the need of a correlational study to describe the existing relationships among all the variables, which has provided the rationale to support the generation of new hypothesis. The results can support future studies designed for clinical proposes, which may help to translate the findings into patient benefit.

## 5. Conclusions

In conclusion, patients with diabetes may have postural adjustments that may be influenced by the interaction among their individual cofactors and sensory deficits, which notably include: sex, BMI, polypharmacy and visual dependence. The novelty of the present study was to explore the interactions among these factors. In addition, the results suggest that the angular VOR at a frequency ~1 Hz and the perception of vertical (i.e., gravity) may have complementary roles in the control of static upright posture. The findings suggest that optimization of rehabilitation programs for adults with diabetes may require an individualized approach. 

## Figures and Tables

**Figure 1 brainsci-12-01489-f001:**
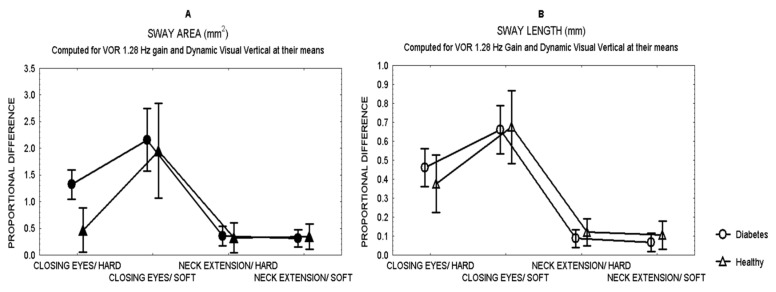
Mean and standard error of the mean of the proportional difference in the area of sway (**A**) and the length of sway (**B**) of patients with type 2 diabetes and healthy subjects, after closing the eyes or 30°neck extension, while standing either on hard or soft surface.

**Figure 2 brainsci-12-01489-f002:**
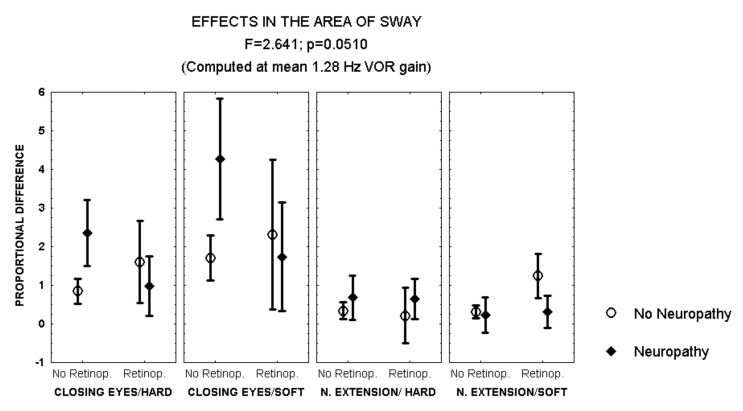
Mean and standard error of the mean of the proportional difference in the area of sway after closing the eyes or extending the neck, while standing either on hard or soft surface, according to the evidence of retinopathy or polyneuropathy. Statistical significance is provided according to the overall result of the MANCOVA, including the variables described in the Figure.

**Figure 3 brainsci-12-01489-f003:**
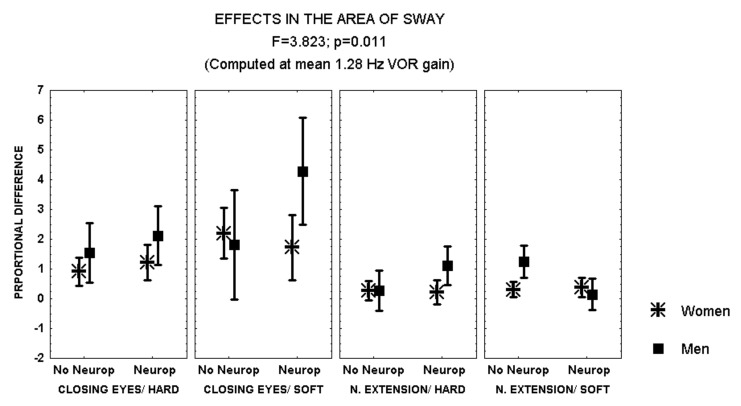
Mean and standard error of the mean of the proportional difference in the area of sway after closing the eyes or extending the neck, while standing either on hard or soft surface, according to sex and the evidence of polyneuropathy. Statistical significance is provided according to the overall result of the analysis by MANCOVA, including the variables described in the Figure.

**Figure 4 brainsci-12-01489-f004:**
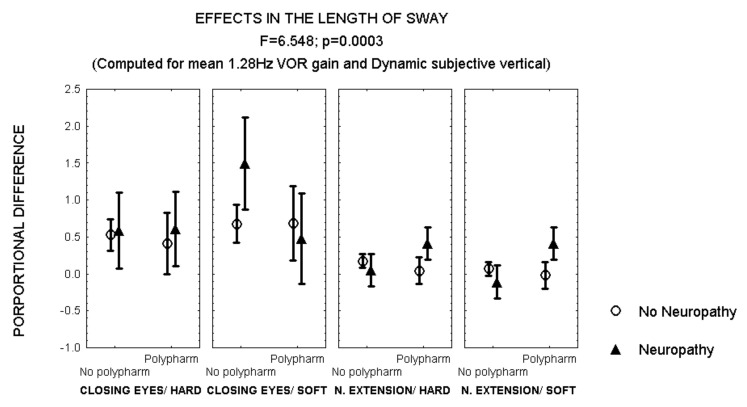
Mean and standard error of the mean of the proportional difference in the length of sway after closing the eyes or extending the neck, while standing either on hard or soft surface, according to polypharmacy and the evidence of polyneuropathy. Statistical significance is provided according to the overall result of the analysis by MANCOVA, including the variables described in the Figure.

**Table 1 brainsci-12-01489-t001:** Sensory challenges included in the study.

Challenge	Recordings
I.	1. Hard surface/open eyes (1st baseline)
	2. Hard surface/closed eyes
II.	3. Hard surface/open eyes (2nd baseline)
	4. Hard surface/open eyes/30° neck extension
III.	5. Soft surface/open eyes (3rd baseline)
	6. Soft surface/closed eyes
IV.	7. Soft surface/open eyes (4th baseline)
	8. Soft surface/open eyes/30° neck extension

**Table 2 brainsci-12-01489-t002:** Results on the evaluations of the participants in the study: 48 patients with diabetes and 21 healthy subjects. Comparisons were performed using “*t*” test.

Test	Diabetes	Healthy	*p*-Value
	*n (%)*	*n (%)*	
Polypharmacy (≥5 drugs per day)	15 (31.2%)	-	NA
Polyneuropathy	18 (37.5%)	-	NA
Retinopathy	17 (35.4%)	-	NA
Polyneuropathy and retinopathy	10 (20.8%)	-	NA
No retinopathy or polyneuropathy	3 (6.2%)	21 (100%)	NA
	*Mean ± Standard deviation*	*Mean ± Standard deviation*	
Quadriceps strength (Kg)	40.1 ± 18.6	58.2 ± 21.6	0.0007
Physical activity score (met/min/week)	2239 ± 2315	3654 ± 2328	0.02
Ankle brachial index	1.00 ± 0.15	0.99 ± 0.03	-
Gain to rotation in the dark at 0.16 Hz	0.47 ± 0.15	0.49 ± 0.13	0.59
Gain to rotation in the dark at 1.28 Hz	0.96 ± 0.12	0.98 ± 0.08	0.48
Static Visual Vertical (°)	−0.37°± 0.83°	−0.02° ± 0.06°	0.058
Dynamic Visual Vertical (absolute average)	4.2°± 1.48°	4.3° ± 0.86°	0.77

NA, non-applicable.

**Table 3 brainsci-12-01489-t003:** Results of the regression analysis on postural sway while standing on hard surface with the eyes open, for the variables included in each model (* *p* < 0.05).

Length of Sway	Estimate	Standard Error	Wald Statistic	*p*-Value
Intercept	5.017	0.415	146.208	< 0.001
Vestibulo-ocular reflex at 1.28 Hz	0.558	0.324	2.954	0.086
Dynamic Visual Vertical (°)	−0.035	0.026	1.782	0.182
Body Mass Index > 30 kg/m^2^	−0.089	0.035	6.272	0.012 *
Sex	−0.094	0.036	6.978	0.008 *
**Area of Sway**				
Intercept	3.869	1.032	14.052	<0.001
Vestibulo-ocular reflex at 1.28 Hz	1.630	0.754	4.674	0.031 *
Dynamic Visual Vertical (°)	−0.108	0.063	2.926	0.087
Quadriceps strength	0.009	0.003	6.564	0.010 *
Body Mass Index > 30 kg/m^2^	−0.165	0.076	4.662	0.031 *

## Data Availability

All data generated or analyzed during this study are included in this article. Further enquiries can be directed to the corresponding author.

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
