# Peer review of "Individual Cofactors and Multisensory Contributions to the Postural Sway of Adults with Diabetes"

_brainsci, 2022, doi:10.3390/brainsci12111489_

Round 1
Reviewer 1 Report
In this paper, Villaseñor-Moreno and colleagues analyze the influence of coindividual factors and sensory inputs on postural stability in a group of adult subjects with diabetes mellitus and a cohort of healthy subjects. The article is generally interesting, quite well-written and the literature cited is adequate. However, i have the below minor points:
- It would be appropriate to specify the unit of measurement utilized to quantify postural sway, both in text and figures.
- The muscles mainly involved in the control of static balance are tibialis anterior, soleus and peroneus longus. Why did you choose to test the quadriceps strength? Please argue that.
- The between-groups statistical comparison should also be reported for age and sex.
- Table 2: Adding the number of subjects with retinopathy, neuropathy and both, the number of diabetic subjects does not match the one indicated in the text. Did some diabetic subject show neither condition? Please clarify that.
- Line 56: could you please cite this review?
Author Response
- IT WOULD BE APPROPRIATE TO SPECIFY THE UNIT OF MEASUREMENT UTILIZED TO QUANTIFY POSTURAL SWAY, BOTH IN TEXT AND FIGURES.
Thank you for the comment. In this revised version of the manuscript, the units are provided in the text and Figure 1; with emphasis on the use of the proportional difference from baseline of each challenge as an “effect” measure (page 3, last paragraph, lines 16-19).
- THE MUSCLES MAINLY INVOLVED IN THE CONTROL OF STATIC BALANCE ARE TIBIALIS ANTERIOR, SOLEUS AND PERONEUS LONGUS. WHY DID YOU CHOOSE TO TEST THE QUADRICEPS STRENGTH? PLEASE ARGUE THAT.
We completely agree with the reviewer. However, Taking into account that muscle strength of both ankle and knee may influence balance control (Kim 2018), the reason to evaluate quadriceps strength as a single measure of muscle strength was two-fold: it has been related to the ability to perform daily life activities /health outcomes in the general population (Hurley 1998) as well as to metabolic control and peripheral neuropathy in patients with diabetes (Strotmeyer 2009); while it is an easy to assess variable that could be included in the complexity of the study protocol of this study.
In order to clarify this issue, the following sections were edited:
- Introduction: a sentence to emphasise the loss of muscle strength/quality/volume in patients with diabetes (page 2, first paragraph, lines 5-6).
- Methods: a comment in the procedures to highlight that, in patients with diabetes, quadriceps strength “has been related to both hyperglycemia and peripheral neuropathy.” (page 3, paragraph 8, lines 3-4).
- Discussion: more information is provided about the muscle groups that are involved in postural balance (page 9, paragraph 4); and the issue is included among the limitations of the study (page 10, first paragraph, lines 4.6).
- Kim SG, et al. Med Sci Monit 2018, 24, 3168-75.
- Hurley MV, et al. Age and Ageing 1998; 27: 55-62.
- Strotmeyer ES. J Am Geriatr Soc 2009, 57, 2004-10
- THE BETWEEN-GROUPS STATISTICAL COMPARISON SHOULD ALSO BE REPORTED FOR AGE AND SEX.
The comparison is now included in the text (page 4, paragraph 3, lines 1-3)
- TABLE 2: ADDING THE NUMBER OF SUBJECTS WITH RETINOPATHY, NEUROPATHY AND BOTH, THE NUMBER OF DIABETIC SUBJECTS DOES NOT MATCH THE ONE INDICATED IN THE TEXT. DID SOME DIABETIC SUBJECT SHOW NEITHER CONDITION? PLEASE CLARIFY THAT.
The subgroup of patients with no retinopathy and no neuropathy is now described in the Table.
- LINE 56: COULD YOU PLEASE CITE THIS REVIEW?
There was a misunderstanding since there are no published articles on the topic. To avoid confusion we have edited the sentence to read: “…, to the best of our knowledge, there are no reports on … “ (page 2, first paragraph, line 10).
We thank the reviewer for the valuable comments.
Reviewer 2 Report
Authors have done a research with significant clinical implications. However, Language of manuscript needs improvement, There are some grammatical errors. Further, methodology needs to be more elaborate. Results can be presented in a better way.
Author Response
AUTHORS HAVE DONE A RESEARCH WITH SIGNIFICANT CLINICAL IMPLICATIONS. HOWEVER, LANGUAGE OF MANUSCRIPT NEEDS IMPROVEMENT, THERE ARE SOME GRAMMATICAL ERRORS
We thank the reviewer for the comments; we have revised the language throughout the manuscript.
FURTHER, METHODOLOGY NEEDS TO BE MORE ELABORATE.
The description of the posturography procedure has been edited for clarity.
RESULTS CAN BE PRESENTED IN A BETTER WAY
We thank the reviewer for the comment. The description of the characteristics of the participants has been edited.
We agree that the results are complex and could be reported in different ways. We made several attempts to describe the results. After showing a variety of options to colleges in the field, all agreed that this was the easiest way to follow the report, in order to provide the reader with the most meaningful findings while avoiding confusion.